# Early Hematoma Evacuation Can Prevent Infectious Complications in Patients with Spontaneous Intracerebral Hemorrhage

**DOI:** 10.3390/jcm14186480

**Published:** 2025-09-14

**Authors:** Daina Kashiwazaki, Kunitaka Maruyama, Shusuke Yamamoto, Emiko Hori, Kyo Noguchi, Satoshi Kuroda

**Affiliations:** 1Departments of Neurosurgery, Graduate School of Medicine and Pharmaceutical Sciences, University of Toyama, 2630 Sugitani, Toyama 930-0194, Japan; kmaru31@med.u-toyama.ac.jp (K.M.); shuyama@med.u-toyama.ac.jp (S.Y.); emihori@med.u-toyama.ac.jp (E.H.); skuroda@med.u-toyama.ac.jp (S.K.); 2Departments of Radiology, Graduate School of Medicine and Pharmaceutical Sciences, University of Toyama, 2630 Sugitani, Toyama 930-0194, Japan; kyo@med.u-toyama.ac.jp

**Keywords:** hematoma evacuation, intracerebral hemorrhage, immunosuppression, infection

## Abstract

**Background/Objectives:** Infections are common complications in patients with spontaneous intracerebral hemorrhage (ICH). This study investigated whether early surgical hematoma evacuation can reduce post-ICH infections and the impact of residual hematomas on infectious complications. **Methods:** Demographic, radiological, and clinical outcome data were collected for 174 patients with spontaneous ICH. The patients were classified according to treatment (Group A, without surgery; Group B, hematoma evacuation with residual hematoma volume ≥10 mL; Group C, hematoma evacuation with residual hematoma volume <10 mL). Kaplan–Meier analysis was used to evaluate infectious complications following ICH, while multivariate logistic regression analysis was used to identify risk factors for infectious complications. **Results:** Groups A, B, and C included 88 (50.6%), 25 (14.4%), and 61 (35.0%) patients, respectively. A total of 68 patients (39.0%) experienced 88 infectious complications, most frequently pneumonia and urinary tract infections. Group C had a significantly lower frequency of infectious complications compared with Groups A and B (*p* = 0.016). The independent risk factors for infectious complications included age, higher National Institutes of Health Stroke Scale score at admission, motor weakness, intraventricular hemorrhage, Group A, and Group B. Patients with infections had longer hospital stays. The frequencies of poor clinical outcomes at one and six months in patients with infection were lower than those in patients without infection (both *p* < 0.01). **Conclusions:** Surgical hematoma evacuation can reduce the risk of post-hemorrhagic stroke infections. Moreover, residual hematoma after surgical evacuation was associated with the risk of cytotoxic effects and subsequent infectious complications.

## 1. Introduction

Spontaneous intracerebral hemorrhage (ICH) accounts for 10–15% of all strokes and is a leading cause of morbidity and mortality [1,2]. Despite numerous clinical trials, no specific treatment has been shown to improve outcomes in patients with spontaneous ICH [3,4,5]. Recent studies have emphasized the critical role of immunoreactions following spontaneous ICH [6,7,8]. ICH profoundly affects systemic immunity related to local immune activation. Systemic immunosuppression occurs soon after early immune activation and is closely associated with infection following ICH. Approximately 30–50% of patients with stroke develop infections, which are a leading cause of life-threatening conditions among extracerebral complications [9,10,11,12,13]. This state of systemic immunosuppression, also known as stroke-induced immunosuppression, can significantly affect ICH [14,15]. Therefore, immunosuppression is a potential therapeutic target.

Among methods for reducing immunosuppression is the evacuation of hematomas associated with ICH. While the main benefit of hematoma evacuation is reducing the mass effect and lifesaving [16], this procedure can also decrease the cytotoxic effects of blood product breakdown [17,18,19]. In the present study, we hypothesized that early surgical hematoma evacuation may reduce immunosuppression to ultimately prevent infectious complications following ICH. Furthermore, the potential contributions of residual hematoma after evacuation to infectious complications owing to post-ICH immunosuppression remain unknown. Therefore, this study aimed to clarify whether surgical hematoma evacuation can reduce the incidence of infectious complications after ICH. Furthermore, we investigated the effect of residual hematomas following hematoma evacuation on infectious complications.

## 2. Materials and Methods

### 2.1. Research Ethics

This cross-sectional study was approved by the Institutional Review Board of our hospital (number 29-145, January 2023). We analyzed a prospective database of patients treated with ICH at our institution. Informed consent was obtained from all the patients or their guardians using the opt-out method. Formal informed consent was not required, in accordance with the ethical standards of the institutional research committees. Instead, the outline of the study was made available to the public on our homepage, and an option for patients to decline inclusion in the research was provided.

### 2.2. Study Design and Population

This study performed a post hoc analysis of a prospective database of patients diagnosed with spontaneous supratentorial ICH with absolute hematoma volume ≥10 mL at our institution between April 2013 and December 2022. The exclusion criteria were (1) age < 20 years upon initial admission; (2) ICH related to trauma, brain tumor, cerebral aneurysm, moyamoya disease, or cerebral arteriovenous malformation; (3) severe liver dysfunction and leukemia, lymphoma, other hematological diseases, or malignant tumors; (4) death within 7 days of ICH onset; and (5) lack of sufficient data. In this study, early hematoma evacuation was defined as surgery performed within 24 h of ICH onset.

### 2.3. Surgical Hematoma Evacuation

Certified neurosurgeons performed all surgical hematoma evacuations. Craniotomy was performed under general anesthesia. The procedure involved scalp incision, cranial drilling, and bone flap creation, followed by opening of the dura mater, sharp Sylvian dissection, and cortical incision to access the hematoma without disrupting the cortical vessels. Hematoma evacuation was performed under the direct vision of a microscope to ensure bleeding points and hemostasis. Surgery was completed by suturing the dura mater, followed by scalp closure to conclude the surgery. To maintain airway patency, a tracheostomy was performed in cases with severely impaired consciousness or poor sputum expectoration.

### 2.4. Clinical and Radiographic Assessments

On admission, demographic and relevant clinical information was obtained from the patient database. ICH was diagnosed using computed tomography (CT) and the volume was calculated using the ABC/2 method [20]. All patients underwent CT angiography (CTA) to exclude conditions including arteriovenous malformations, moyamoya disease, and aneurysms. Head CT was performed immediately after surgery to assess hematoma clearance and detect any rebleeding or residual hematoma.

The patients were divided into subgroups based on whether the hematoma was evacuated and whether a residual hematoma was present. Group A included patients without surgical hematoma evacuation. Group B included patients who underwent surgical hematoma evacuation with residual hematoma volume ≥10 mL. Finally, Group C included patients who underwent surgical hematoma evacuation and with residual hematoma volume <10 mL. A typical case from each group is shown in Figure 1.

The therapeutic interventions included respiratory failure requiring mechanical ventilation, external ventricular drain (EVD) placement, tracheostomy, and central line placement.

Infectious complications were defined as any type of infection diagnosed during hospitalization, characterized by fever (axillary temperature ≥ 37.5 °C) in patients with positive microbiological cultures.

### 2.5. Blood Sampling

Serological data were acquired by collecting blood from the peripheral vein of each patient at admission and one, 6–8, and 13–16 days after admission. Serum C-reactive protein (CRP) and neutrophil-to-lymphocyte ratio (NLR) were measured in the obtained blood samples as markers of inflammatory response [21,22,23]. The NLR was defined as the absolute neutrophil count divided by the absolute lymphocyte count.

### 2.6. Clinical Outcomes

The clinical data were collected from a database. Stroke severity was assessed using National Institutes of Health Stroke Scale (NIHSS) scores at admission (baseline). All outcome variables at one and six months after onset were dichotomized according to functional outcome (modified Rankin Scale [mRS]) as good (0–3) or poor (4–6). All outcome data were adjudicated by at least two attending neurosurgeons at weekly clinical conferences.

### 2.7. Statistical Analysis

Normally distributed data are presented as means ± standard deviation (SD). Categorical variables are presented as frequencies and percentages. Statistical significance was set at *p* < 0.05. significant. Continuous data were analyzed using the chi-square test, one-factor analysis of variance (ANOVA) followed by post hoc Bonferroni test, or Mann–Whitney U-test, as appropriate. The cumulative incidence of infectious complications was estimated using Kaplan–Meier survival analysis. To compare the risk of infectious complications, the patients were further subdivided for logistic analysis into those with pneumonia, urinary tract infection (UTI), or any type of infection. To compare the risk factors for poor outcomes, we performed a logistic analysis.

## 3. Results

### 3.1. Descriptive Statistics and Comparisons Among Patient Groups

There were 243 patients who diagnosed as ICH during study period. Of them, 69 patients (24.8%) were excluded by above mentioned exclusion criteria. This study included 86 patients (49.4%) who underwent surgical hematoma evacuation (Groups B and C) and 88 patients (50.5%) who did not undergo hematoma evacuation (Group A). Among them, postoperative CT revealed residual hematoma ≥10 mL in 25 patients (28.4%, Group B) and <10 mL in 61 patients (71.6%, Group C). The demographic and radiological findings, infectious complications, and clinical outcomes of each group are summarized in Table 1.

The initial absolute hematoma volumes were 26.4 ± 12.2, 42.2 ± 9.9, and 43.2 ± 9.1 mL in Groups A, B, and C, respectively (*p* < 0.01). The residual hematoma volumes following hematoma evacuation were 16.8 ± 4.5 mL in Group B and 4.8 ± 3.1 mL in Group C (*p* < 0.01). Early surgical hematoma evacuation (<24 h) was performed in 22 (88.0%) and 50 (82.0%) patients in Groups B and C, respectively.

NLR at admission did not differ significantly among the three groups (*p* = 0.51, Group A: 3.66 ± 0.98; Group B: 3.76 ± 1.33; Group C: 3.89 ± 1.27). However, NLR after one and 7–9 days in Group C were significantly lower than those of Groups A and B (one day: *p* = 0.04, Group A:3.98 ± 0.98, Group B: 4.66 ± 1.33, Group C: 3.87 ± 1.27; 7–9 days: *p* = 0.01, Group A:4.62 ± 1.48, Group B: 5.38 ± 1.55, Group C: 3.88 ± 1.37). Finally, the NLR after 14–16 days did not differ significantly among the three groups (*p* = 0.35, Group A: 2.21 ± 0.78, Group B: 2.62 ± 0.95, Group C: 2.11 ± 0.84).

Similarly, CRP levels at admission did not differ significantly among the three groups (*p* = 0.23, Group A:1.96 ± 1.12, Group B: 2.05 ± 1.31 Group C: 1.05 ± 1.27). CRP levels after one day and 7–9 days in Group C were significantly lower than those of Groups A and B (one day: *p* = 0.02, Group A:3.25 ± 0.98, Group B: 4.25 ± 1.33, Group C: 2.89 ± 1.27; 7–9 days: *p* < 0.01, Group A:4.34 ± 1.58, Group B: 5.05 ± 1.36 Group C: 2.39 ± 1.31). The NLR and CRP levels are presented in Figure 2.

Severe neurological status at admission (NIHSS score > 15) was observed in 36 patients (40.9%) in Group A, 15 patients (60.0%) in Group B, and 33 patients (54.1%) in Group C (*p* = 0.04).

### 3.2. Infectious Complications

In total, 68 patients (39.0%) developed 88 infectious complications during the inpatient postsurgical recovery period. Eighteen (26.5%) patients had multiple organ system infections. The most frequent infection was pneumonia (*n* = 40, 45.5%), followed by UTI (*n* = 32, 36.4%), sepsis (*n* = 7, 8.0%), meningitis (*n* = 5, 5.7%), and biliary infection (*n* = 4, 4.5%). Similarly to previous studies [9,24], 78.4% of infectious complications were first identified as culture-positive 10 days after onset (Figure 3).

Numerous organisms were cultured from the patient samples. Most frequently, *Staphylococcus* (22.5%) and *Streptococcus* (17.5%) were detected in sputum cultures, whereas *Klebsiella* (18.8%) and *Escherichia coli* (15.6%) were detected in urine cultures (Figure 4).

The frequency of infectious complications varied widely among the three groups. Forty-eight infectious complications were observed in 40 patients (45.5%) in Group A, 20 in 13 patients (52.0%) in Group B, and 18 in 15 patients (24.6%) in Group C. Infectious complications were less frequent in Group C than in Groups A and B (*p* = 0.01). Kaplan–Meier analysis demonstrated a significantly lower frequency of infectious complications in Group C than in Groups A or B (*p* = 0.016) (Figure 5).

In Group C (*n* = 61), 15 patients (24.6%) developed an infection following hematoma evacuation. Of these, nine and six patients were treated with early and late hematoma evacuation, respectively. Patients who underwent early surgical hematoma evacuation had a lower incidence of infection than those who underwent late surgical hematoma evacuation (*p* = 0.02). However, Group B showed no significant difference between early and late surgical evacuations (*p* = 0.31).

Multivariate analysis revealed that patients with perioperative infection were significantly older (*p* = 0.01, odds ratio [OR]:1.61, 95% confidence interval [CI] 1.19–4.56), while sex showed no significant association (*p* = 0.18). Higher NIHSS (NIHSS > 15, *p* < 0.01, OR: 1.22, 95%CI 1.02–1.98), lower Glasgow Coma Score (GCS < 12, *p* < 0.01, OR: 1.3, 95%CI 1.05–2.04), and motor weakness (*p* = 0.03, OR: 1.11, 95%CI 1.02–1.31) on admission was associated with a higher incidence of postoperative infection. Patients with infection were also more likely to present with intraventricular hemorrhage (*p* = 0.02, OR: 1.09, 95%CI 1.03–1.32). Furthermore, surgical hematoma evacuation with a residual hematoma volume <10 mL occurred less frequently in patients with infectious complications (*p* = 0.01, OR: 0.85, 95%CI 0.79–0.92). The results of the multivariate analysis to evaluate the independent risk factors for infection, pneumonia, and UTI are presented in Table 2.

In this study, the clinical outcome of mRS score of 4–6 was achieved by 51 (58.0%), 15 (60.0%), and 30 (49.2%) patients in Groups A, B, and C, respectively, a one month after ICH onset, and 31 (35.2%), nine (36.0%), and 20 (32.8%) at six months. The clinical outcomes at one and six months did not differ significantly among the three groups (*p* = 0.21 and *p* = 0.11, respectively). The patient outcomes at the one- and six-month follow-ups were worse for patients with infections than for those without infections (both *p* < 0.01) (Figure 6). The hospital stay durations were 23.1 ± 9.1, 24.6 ± 10.5, and 19.2 ± 8.9 days in Groups A, B, and C, respectively (*p* = 0.03), and was shorter in Group C than in Groups A and B (*p* < 0.01).

## 4. Discussion

Infections are common systemic complications in patients treated for spontaneous ICH and can lead to an extended length of hospital stay and poor functional outcomes. The main objective of this study was to investigate the effect of hematoma evacuation, which reduces the cytotoxic effects of blood product breakdown, on immunosuppression and subsequent infectious complications in this patient population. Our data demonstrated a lower frequency of postoperative infection in patients who underwent hematoma evacuation with residual hematoma volume <10 mL compared with residual hematoma ≥10 mL among patients with ICH who underwent surgical treatment. Therefore, residual hematoma should be minimized to reduce cytotoxic effects and promote immunosuppression. The superiority of clinical outcomes in the surgical group could not be determined in this study because the initial neurological severity was more severe in Groups B and C than in Group A. However, our study results suggest poor clinical outcomes in patients with infectious complications.

### 4.1. Benefits of Hematoma Evacuation

The chief benefits of surgical hematoma evacuation are that it reduces the mass effect, avoids midline shift, and improves cerebral perfusion by decreasing intracranial pressure. Second, hematoma removal has potential advantages in ameliorating the risk of secondary brain damage by reducing the cytotoxic effects of blood products, peri-hematoma edema, and ischemia due to the mass effect [16,17]. The results of the present study suggest that surgical hematoma evacuation can reduce the frequency of infectious complications, thereby contributing to a shorter hospital stay. Therefore, hematoma evacuation can reduce secondary brain injury. The mechanism of infection in the peripheral organs after stroke includes the activation of the sympathetic nervous system, hypothalamus–pituitary–adrenal axis, and immune system, leading to a series of systemic events and, finally, to the injury of various peripheral organs [25]. However, the role of open surgery in treating spontaneous ICH remains controversial. While the Surgical Trial in Intracerebral Hemorrhage (STICH), a large, multicenter, randomized clinical trial, compared the benefits of early hematoma drainage with initial conservative management [26], it did not report the functional outcomes or mortality benefits of early hematoma evacuation. Consequently, a second study (STICH II) was performed to test the hypothesis that patients with superficial hematomas within 1 cm of the cortical surface could benefit from early hematoma removal (early surgery versus initial conservative treatment in patients with spontaneous lobar intracerebral hemorrhage) [27]. This trial failed to demonstrate the benefits of early hematoma evacuation. However, caution should be exercised in interpreting these findings because these trials excluded patients considered to benefit from surgery, such as those with high intracranial pressure and midline shift. Large hemorrhages may lead to life-threatening cerebral or brainstem herniation that may require life-saving emergency surgical evacuation. In such clinical scenarios, the best medical management is probably not equipoised with surgery, which prevents the inclusion of these patients in a randomized clinical trial. Thus, the ideal candidates and the optimal timing of surgery are essential questions that have not yet been determined; however, our data may help resolve these issues.

### 4.2. Optimal Timing of Hematoma Evacuation to Prevent Infectious Complications

Infections are common complications in patients treated for spontaneous ICH and can lead to an extended length of stay and poor functional outcomes [12,28]. In the present study, nearly half of all patients with ICH developed infectious complications. The results of our study showed that the benefit of reducing postoperative infections was limited to early hematoma evacuation. The CRP level and NLR also supported these results, which increased and differed significantly among the three groups from one day after surgery. Pradilla et al. suggested that early hematoma evacuation (<24 h) resulted in better functional outcomes, observing a lower incidence of lung infection in the early surgical hematoma evacuation group compared with the control group. Previous reports observed peripheral immune activation characterized by lymphocytopenia, decreased T lymphocyte count, and delayed recovery of T lymphocytes 24 h after onset, which were associated with infectious complications following ICH [10,14,15,17,29]. Therefore, early hematoma evacuation (<24 h from ICH onset) is optimal to reduce infectious complications. Notably, the patients in the present study underwent surgery within the first 24 h after stroke.

### 4.3. Immunosuppression Related to ICH

Rapid reaction of the sympathetic nervous system/hypothalamus–pituitary–adrenal (SNS/HPA) axis following ischemic and hemorrhagic stroke is thought to be the reciprocal relation between the CNS and the peripheral immune system (27753158). It is known SNS/HPA system activation may have effects on post-hemorrhagic stroke immune deficiency and infection risks. Zhang et al. suggested lymphocytopenia was present and sustained up to day 14 post-ICH. Monocyte counts also appear to be associated with ICH outcome (28877956). These reports supported our results including NLR and CRP change following ICH and surgical hematoma evacuation. In addition to neurological function and hematoma size, higher NLR in ICH is also associated with infectious complications (28419988). This immunosuppression cascade might be a therapeutic target. It is known stroke activates the SNS, inducing lymphocyte apoptosis and lymphoid organ atrophy (26303850, 22129259). Some studies suggested blockade of adrenergic signal by the beta-blocker prevents the decrease in lymphocytes and reverses immunosuppression, which resulted in reducing bacterial infection in animal stroke models (27694934, 2194193). This treatment is expected to be effective in humans.

### 4.4. Limitations

This study has several limitations. First, this was a single-center study. Second, the sample size was insufficient to include a significant number of patients with certain infection types such as sepsis and meningitis. A larger multi-institutional cohort study may sufficiently capture low-frequency events. Third, the findings may not be generalizable to other institutions that use different infection control protocols. A multi-institutional prospective patient cohort is required to validate these results across settings. Fourth, in this study, multiple surgeons performed hematoma evacuation. Generally, they performed under same surgical manner, surgical technique variability including skill variability might influence residual hematoma volume and surgical outcome. Fifth, this study included various clinical severity; no doubt, clinical disease severity is itself a strong predictor of clinical outcome and infection rate. Further study with prospective study including ICH patients in similar clinical severity is warranted. Finally, infection was defined as fever (axillary temperature ≥ 37.5 °C) and positive microbiological cultures. However, this definition may underestimate infectious complications.

## 5. Conclusions

In conclusion, infectious complications are common in patients with spontaneous ICH. The results of this study demonstrated that hematoma evacuation can reduce the risk of infectious complications post-hemorrhagic stroke. Moreover, residual hematoma after surgical evacuation was associated with the risk of cytotoxic effects and subsequent infectious complications. Overall, our findings provide clinicians with useful information regarding the impact of individual infection types on patient complications and recovery from spontaneous ICH.

## Figures and Tables

**Figure 1 jcm-14-06480-f001:**
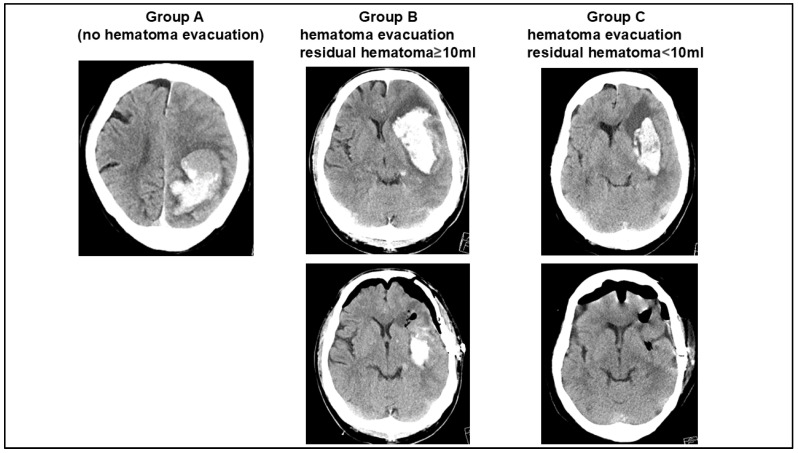
Illustrative cases for each patient group. Group A: patients without hematoma evacuation. Group B: patients receiving hematoma evacuation with ≥10 mL residual hematoma volume. Group C: patients receiving hematoma evacuation with <10 mL residual hematoma volume.

**Figure 2 jcm-14-06480-f002:**
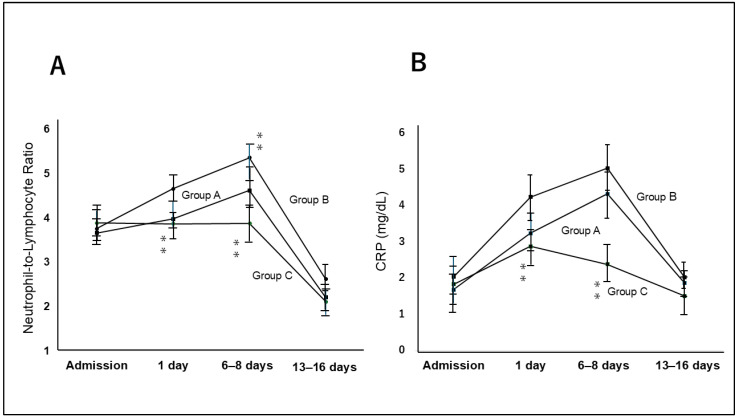
Bar graph showed serum neutrophil to lymphocyte ratio (NLR) (**A**) and C-reactive protein (CRP) levels (**B**) in each patient at admission and one, 7–8, and 13–16 days from admission. NRL and CRP levels on the day of admission and one and 6–8 days from admission, were significantly lower in Group C compared with Groups A and B. NRL and CRP values were highest on 6–8 days from admission. There were no significantly difference among 3 groups on admission day and 13–16 days from admission. ** means *p* < 0.01.

**Figure 3 jcm-14-06480-f003:**
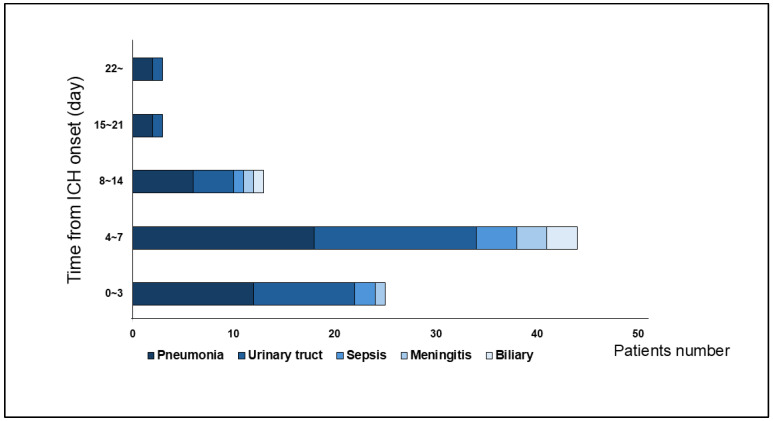
This bra graph showed number and timing of infectious complications from admission. Pneumonia and urinary tract infections were the most frequent. Most infections occurred within 7 days from admission.

**Figure 4 jcm-14-06480-f004:**
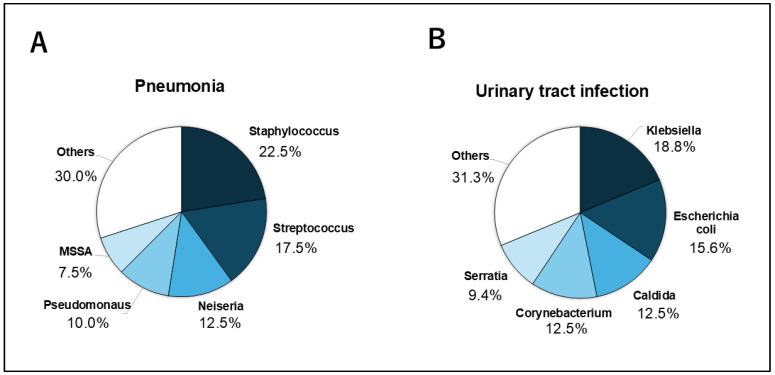
This pie graph showed causative bacteria in pneumonia and urinary tract infection. (**A**) *Staphylococcus* (22.5%) following *Streptococcus* (17.5%) and *Neiseria* (12.5%) were the most frequent bacteria detected in pneumonia. (**B**) *Klebsiella* (18.8%) following Escherichia (15.6%) and Caldlda (12.5%) in urinary tract infection.

**Figure 5 jcm-14-06480-f005:**
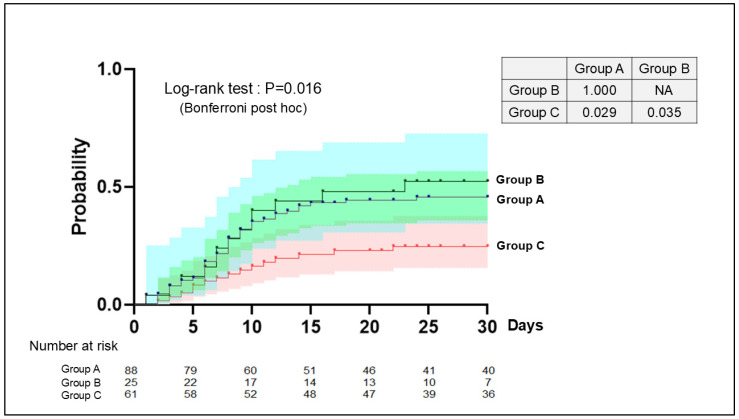
Kaplan–Meier plot of infections in each group. Kaplan–Meier analysis demonstrated a significantly lower frequency of infectious complications in Group C than in Groups A or B (*p* = 0.016). The post hoc analysis showed the incidence of infectious complication were significantly difference between Group A and C (*p* = 0.029) and Group B and C (*p* = 0.035). There were not significant differences between Group A and B (*p* = 1.000).

**Figure 6 jcm-14-06480-f006:**
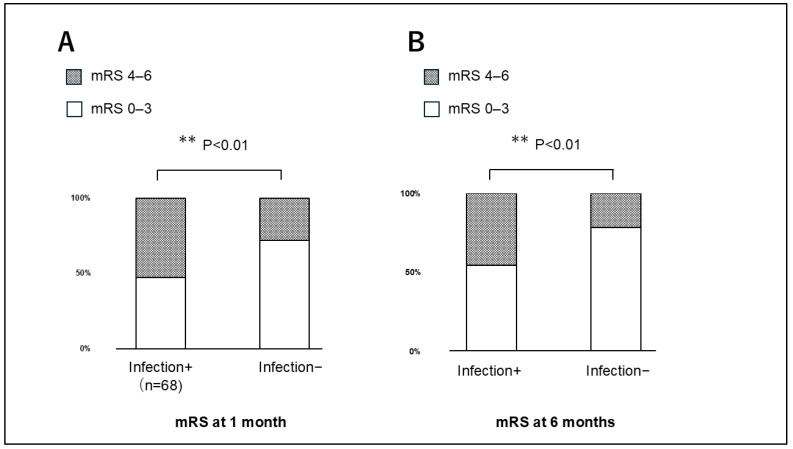
Patient outcomes at the (**A**) one- and (**B**) six-month follow-ups, showing worse outcomes among patients with infections compared with those without infections (both *p* < 0.01).

**Table 1 jcm-14-06480-t001:** Summary of demographic, radiological, and clinical data.

	Group A	Group B	Group C	*p*
	without hematoma evacuation	hematoma evacuation with residual hematoma ≥ 10 mL	hematoma evacuation with residual hematoma < 10 mL	
**Number of patients**	88	25	61	NA
**Baseline characteristics**				
age(years)	68.8 ± 11.3	65.3 ± 12.1	64.4 ± 12.5	0.43
gender, male	56 (63.6%)	16 (64.0%)	39 (63.9%)	
**Comorbidities**				
hypertension	85 (96.6%)	23 (92.0%)	58 (95.1%)	0.38
dyslipidemia	29 (33.0%)	10 (40.0%)	18 (29.5%)	0.21
diabetic mellitus	25 (28.4%)	9 (36.0%)	17 (27.9%)	0.44
coronary artery disease	11 (12.5%)	4 (16.0%)	9 (14.8%)	0.54
renal failure	10 (11.4%)	4 (16.0%)	7 (11.5%)	0.27
**Surgery**				
surgery in <24 h	NA	22 (88.0%)	50 (82.0%)	NA
ventricle drain	8 (9.1%)	4 (16.0%)	8 (13.1%)	0.17
tracheostomy	9 (10.2%)	5 (20.0%)	8 (13.1%)	
**Radiological findings**				
hematoma volume (mL)	26.4 ± 5.2	42.2 ± 9.9	43.2 ± 9.1	<0.01
residual hematoma (mL)	NA	16.8 ± 4.5	4.8 ± 3.1	<0.01
IVH	19 (21.6%)	5 (20.0%)	11 (18.0%)	0.22

**Table 2 jcm-14-06480-t002:** This table showed multivariate analysis to predict each infection type.

	All Infection			Pneumonia			Urinary Tract Infection		
	OR	95% CI	*p*	OR	95% CI	*p*	OR	95% CI	*p*
**Demographic**									
Age	**1.61**	**1.19–4.56**	**0.02**	**1.52**	**1.15–3.86**	**0.01**	**1.65**	**1.18–4.02**	**0.02**
Gender (female)	1.02	0.94–1.88	0.18	0.95	0.90–1.95	0.13	**1.22**	**1.12–4.15**	**0.02**
Diabetec mellitus	1.04	0.90–1.45	0.32	1.03	0.92–1.35	0.21	1.02	0.90–1.45	0.28
Dyslipidemia	1.01	0.88–1.35	0.56	1.03	0.94–1.18	0.31	1.04	0.90–1.15	0.41
History of ischemic storke	1.02	0.95–1.21	0.09	1.01	0.92–1.12	0.15	1.02	0.91–1.05	0.17
History of hemorrhagic stroke	1.01	1.10–1.35	0.22	0.99	0.92–1.10	0.26	1.02	0.94–1.05	0.27
**Neurological**									
NIHSS > 15	**1.22**	**1.02–1.98**	**<0.01**	**1.35**	**1.04–2.33**	**<0.01**	**1.16**	**1.01–1.84**	**0.01**
GCS < 12	**1.3**	**1.05–2.04**	**<0.01**	**1.53**	**1.08–2.08**	**<0.01**	**1.21**	**1.02–1.55**	**0.02**
motor weakness	**1.11**	**1.02–1.31**	**0.03**	**1.12**	**1.03–1.21**	**0.02**	**1.09**	**1.00–1.17**	**0.04**
motor aphasia	1.04	0.92–1.32	0.38	1.03	0.94–1.27	0.44	1.04	0.92–1.18	0.17
sensory aphasia	1.02	0.88–1.03	0.21	1.03	0.87–1.33	0.46	0.99	0.82–1.22	0.25
**Radiological**									
intraventrivle hemorrhage	**1.09**	**1.03–1.32**	**0.02**	**1.11**	**1.05–1.23**	**0.02**	**1.07**	**1.03–1.25**	**0.03**
Hemorrhage									
cortical	1.03	0.91–1.09	0.21	1.01	0.92–1.11	0.31	0.98	0.90–1.09	0.28
putamen	0.95	0.84–1.06	0.18	0.98	0.92–1.06	0.31	0.94	0.81–1.19	0.27
thalamus	1.02	0.94–1.10	0.38	1.02	0.88–1.12	0.41	1.01	0.9–1.12	0.36
others	1.03	0.91–1.21	0.45	1.05	0.89–1.12	0.17	1.02	0.96–1.09	0.22
**Medical**									
mechanical ventilation > 24 h	0.98	0.88–1.20	0.36	0.97	0.86–1.18	0.28	0.99	0.91–1.08	0.46
EVD use	1.05	0.94–1.14	0.15	1.06	0.95–1.08	0.21	1.04	0.91–1.10	0.3
hematoma evacuation	0.96	0.92–0.1.01	0.08	0.95	0.91–1.01	0.07	0.97	0.94–1.04	0.1
hematoma evacuation with residual <10 mL	**0.85**	**0.79–0.92**	**0.01**	**0.82**	**0.75–0.90**	**0.01**	**0.88**	**0.82–0.97**	**0.02**

## Data Availability

The data analyzed in the current study are available from the corresponding author upon reasonable request.

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
