# Peer review of "Early Hematoma Evacuation Can Prevent Infectious Complications in Patients with Spontaneous Intracerebral Hemorrhage"

_jcm, 2025, doi:10.3390/jcm14186480_

Round 1
Reviewer 1 Report
Comments and Suggestions for Authors
This is a well-structured and clinically meaningful study addressing the relationship between hematoma evacuation and prevention of infectious complications in spontaneous intracerebral hemorrhage (ICH). The authors present a relatively large single-center cohort with detailed clinical, radiological, and laboratory data. The finding that early and more complete hematoma evacuation reduces infection rates is novel, clinically relevant, and has potential to influence neurosurgical decision-making in this difficult patient population. The methodology is sound, the statistical approach appropriate, and the results are clearly presented.
That said, there are some points that need to be discussed :
1. Clarify patient selection and exclusions: While the criteria are generally well defined, it would be helpful to emphasize how many patients were excluded for lack of sufficient data and whether this could bias the results.
2. Strengthen the discussion of mechanisms: The link between residual hematoma, systemic immunosuppression, and infection is compelling. The authors could expand slightly on the biological rationale and cite more recent immunological studies in ICH.
3. Improve figures and tables: Figure 2 (NLR and CRP) and Figure 5 (Kaplan–Meier curve) are central to the findings, but the legends could be more detailed, table 1 is too desne.
4. Outcome reporting: Although the infection outcomes are clear, the authors may wish to emphasize in the discussion why functional outcomes did not differ significantly among groups, despite reduced infections.
Limitations: The limitations section is well written, but a brief note on how surgical technique variability (even if all procedures were performed by certified neurosurgeons) might influence residual hematoma volume would strengthen transparency.
Author Response
Reviewer 1
Comment
This is a well-structured and clinically meaningful study addressing the relationship between hematoma evacuation and prevention of infectious complications in spontaneous intracerebral hemorrhage (ICH). The authors present a relatively large single-center cohort with detailed clinical, radiological, and laboratory data. The finding that early and more complete hematoma evacuation reduces infection rates is novel, clinically relevant, and has potential to influence neurosurgical decision-making in this difficult patient population. The methodology is sound, the statistical approach appropriate, and the results are clearly presented.
Response: Thank you very much for providing important comments. We are thankful for the time and energy you expended. Our responses to the referees’ comments are as follow.
Comment 1. Clarify patient selection and exclusions: While the criteria are generally well defined, it would be helpful to emphasize how many patients were excluded for lack of sufficient data and whether this could bias the results.
Response to reviewer 1.
We strongly appreciate the reviewer's comment on this point. We agree that this point requires clarification, and have added the followed sentences in result section.
Added to text. (line 138-139)
There were 243 patients who diagnosed as ICH during study period. Of them, 69 patients (24.8%) were excluded by above mentioned exclusion criteria.
Comment 2. Strengthen the discussion of mechanisms: The link between residual hematoma, systemic immunosuppression, and infection is compelling. The authors could expand slightly on the biological rationale and cite more recent immunological studies in ICH.
Response to reviewer
We wish to express our deep appreciation to the reviewer for his insightful comment on this point. In revised manuscript, therefore, we added one paragraph about the relationship between hemorrhage-induced immunosuppression and interpret your results in discussion segment.
Added to text.
4.3. Immunosuppression related to ICH (line 292-306)
Rapid reaction of the sympathetic nervous system/hypothalamus-pituitary-adrenal (SNS/HPA) axis following ischemic and hemorrhagic stroke is thought to be the reciprocal relation between the CNS and the peripheral immune system (27753158). It is known SNS/HPA system activation may have effects on post-hemorrhagic stroke immune deficiency and infection risks. Zhang et al. suggested lymphocytopenia was present and sustained up to day 14 post-ICH. monocyte counts also appear to be associated with ICH outcome (28877956). These reports supported our results including NLR and CRP change following ICH and surgical hematoma evacuation. In addition to neurological function and hematoma size, higher NLR in ICH is also associated with infectious complications (28419988). This immunosuppression cascade might be a therapeutic target. It is known stroke activates the SNS, inducing lymphocyte apoptosis and lymphoid organ atrophy (26303850, 22129259). Some studies suggested blockade of adrenergic signal by the beta-blocker prevents the decrease of lymphocytes and reverses immunosuppression, which resulted in reducing bacterial infection in animal stroke models (27694934, 2194193). This treatment is expected to be effective in humans.
- Improve figures and tables: Figure 2 (NLR and CRP) and Figure 5 (Kaplan–Meier curve) are central to the findings, but the legends could be more detailed, table 1 is too desne.
Response to reviewer: We completely agree with reviewer’s advice, in revised manuscript therefore, we added detailed explanation in each figure legends. Further table 1 was briefly revised.
Change to text: (in each figure legends)
Figure 2. Bar graph showed serum neutrophil to lymphocyte ratio (NLR) and C-reactive protein (CRP) levels in each patient at admission and one, 7–8, and 13–16 days from admission. NRL and CRP levels on the day of admission and one and 6–8 days from admission, were significantly lower in Group C compared with Groups A and B. NRL and CRP values were highest on 6-8 days from admission. There were no significantly difference among 3 groups on admission day and 13-16days from admission. **means P<0.01.
Figure 3. This bra graph showed number and timing of infectious complications from admission. Pneumonia and urinary tract infections were the most frequent. Most infections occurred within 7 days from admission.
Figure 4. This pie graph showed causative bacteria in pneumonia and urinary tract infection. Staphylococcus (22.5%) following Streptococcus (17.5%) and Neiseria (12.5%) were the most frequent bacteria detected in pneumonia. Klebsiella (18.8%) following Escherichia (15.6%) and Caldlda (12.5%) in urinary tract infection.
Figure 5. Kaplan–Meier plot of infections in each group. Kaplan–Meier analysis demonstrated a significantly lower frequency of infectious complications in Group C than in Groups A or B (P=0.016). The post-hoc analysis showed the incidence of infectious complication were significantly difference between Group A and C (P=0.029) and Group B and C (P=0.035). There were not significantly difference between Group A and B (P=1.000).
4 Comment: Limitations: The limitations section is well written, but a brief note on how surgical technique variability (even if all procedures were performed by certified neurosurgeons) might influence residual hematoma volume would strengthen transparency.
Response to reviewer
We wish to express our deep appreciation to the reviewer for his insightful comment on this point. To make this point clearer, we have added the following text in limitation section.
Added to text (line 314-316)
Fourth, in this study, multiple surgeons performed hematoma evacuation. Generally, they performed under same surgical manner, surgical technique variability including skill variability might influence residual hematoma volume and surgical outcome.
Reviewer 2 Report
Comments and Suggestions for Authors
The study expands the benefit of hematoma evacuation from the traditional goal of "mass effect reduction" to "alleviating immunosuppression and infection," offering a unique viewpoint and a new theoretical basis for ICH treatment.
There are some limitations and recommendations.
- The baseline disease severity significantly differs among the three groups. This complicates the direct comparison of infection rates and outcomes, as the initial disease severity is itself a strong predictor of both.
- Please supplement the results with specific P-values in Table 1.
- Relying solely on "fever + positive microbiological culture" may underestimate the true infection rate. It is recommended to define infections using established criteria (such as the CDC criteria for nosocomial infections) to improve the comparability and generalizability of the results. At a minimum, this should be acknowledged as a limitation.
- The Discussion should more deeply engage with the literature on hemorrhage-induced immunosuppression and interpret your results (NLR/CRP changes) within this theoretical framework. This would strengthen the proposed causal chain: "hematoma evacuation → reduced immunosuppression → fewer infections."
Author Response
Reviewer 2
The study expands the benefit of hematoma evacuation from the traditional goal of "mass effect reduction" to "alleviating immunosuppression and infection," offering a unique viewpoint and a new theoretical basis for ICH treatment.
There are some limitations and recommendations.
Comment 1 The baseline disease severity significantly differs among the three groups. This complicates the direct comparison of infection rates and outcomes, as the initial disease severity is itself a strong predictor of both.
Response to reviewer: We completely agree with reviewer’s advice, in revised manuscript therefore, we added followed limitations to explain disease severity itself is predictor of clinical outcome and infectious complications.
Added to text (316-319)
Fifth, this study included various clinical severity, no doubtly clinical disease severity is itself a strong predictor of clinical outcome and infection rate. Further study with prospective study including ICH patients in similar clinical severity should be warrant.
Comment 2. Please supplement the results with specific P-values in Table 1.
Response to reviewer: We completely agree with reviewer’s opinion. In revised manuscript, therefore, we added P value in Table 1.
Comment 3 Relying solely on "fever + positive microbiological culture" may underestimate the true infection rate. It is recommended to define infections using established criteria (such as the CDC criteria for nosocomial infections) to improve the comparability and generalizability of the results. At a minimum, this should be acknowledged as a limitation.
Response to reviewer
We wish to express our deep appreciation to the reviewer for his insightful comment on this point. Unfortunately, we could not evaluate using detailed definition like CDC criteria for nosocomial infection, because our protocol in this study did not match to these infection criteria. We added this point as limitations.
Added to text (line 319-321)
Finally, infection was defined as fever (axillary temperature ≥37.5°C) and positive microbiological cultures. However, this definition may underestimate infectious complications.
Comment 4 The Discussion should more deeply engage with the literature on hemorrhage-induced immunosuppression and interpret your results (NLR/CRP changes) within this theoretical framework. This would strengthen the proposed causal chain: "hematoma evacuation → reduced immunosuppression → fewer infections."
Response to reviewer
We wish to express our deep appreciation to the reviewer for his insightful comment on this point. In revised manuscript, therefore, we added one paragraph about the relationship between hemorrhage-induced immunosuppression and interpret your results in discussion segment.
Added to text.
4.3. Immunosuppression related to ICH (line 292-306)
Rapid reaction of the sympathetic nervous system/hypothalamus-pituitary-adrenal (SNS/HPA) axis following ischemic and hemorrhagic stroke is thought to be the reciprocal relation between the CNS and the peripheral immune system (27753158). It is known SNS/HPA system activation may have effects on post-hemorrhagic stroke immune deficiency and infection risks. Zhang et al. suggested lymphocytopenia was present and sustained up to day 14 post-ICH. monocyte counts also appear to be associated with ICH outcome (28877956). These reports supported our results including NLR and CRP change following ICH and surgical hematoma evacuation. In addition to neurological function and hematoma size, higher NLR in ICH is also associated with infectious complications (28419988). This immunosuppression cascade might be a therapeutic target. It is known stroke activates the SNS, inducing lymphocyte apoptosis and lymphoid organ atrophy (26303850, 22129259). Some studies suggested blockade of adrenergic signal by the beta-blocker prevents the decrease of lymphocytes and reverses immunosuppression, which resulted in reducing bacterial infection in animal stroke models (27694934, 2194193). This treatment is expected to be effective in humans.
Round 2
Reviewer 2 Report
Comments and Suggestions for Authors
Thanks for your respond, I have no other questions.